# Radiation Overuse in Intensive Care Units

Chiara Zanon [1,*], Costanza Bini [1], Alessandro Toniolo [1], Tommaso Benetti [2] and Emilio Quaia [1]

1   Department of Radiology, University Hospital of Padua, 35128 Padua, Italy
2   Department of Medicine, University of Padua, 35128 Padua, Italy
*   Correspondence: chiara.zanon@unipd.it

**Abstract:** Radiological imaging is essential for acute patient management in Intensive Care Units (ICUs); however, it introduces the risk of exposure to ionizing radiation. This review synthesizes research on radiation exposure in ICU settings, highlighting its rise during the COVID-19 pandemic and the rise in repetitive imaging. Our analysis extends to radiation safety thresholds, revealing that some ICU patients exceed the diagnostic reference level, emphasizing the need to balance diagnostic utility against potential long-term risks, such as cancer. Prospective studies have demonstrated an increase in the median cumulative effective dose in patients with trauma over time, prompting calls for improved awareness and monitoring. Moreover, innovative dose-reduction strategies and optimized protocols, especially in neuro-ICUs, promise to mitigate these risks. This review highlights the essential but risky role of radiological imaging in critical care. It advocates for rigorous radiation management protocols to safeguard patient health while ensuring the continuity of high-quality medical care.

**Keywords:** radiological imaging; radiation exposure; intensive care units; COVID-19; optimization strategies; cancer risk; cumulative effective dose

## 1. Introduction

Radiological imaging has become an indispensable tool in the modern healthcare landscape, particularly in Intensive Care Units (ICUs). These specialized units are dedicated to the care of patients with critical conditions, where timely and accurate diagnoses are often a matter of life and death. Radiological imaging, including radiography and computed tomography (CT), serves as a lifeline for physicians in these high-stress settings, offering crucial insights into patient conditions and aiding continuous health monitoring [1–3].

However, this invaluable resource comes with a consequential caveat: the potential risk of ionizing radiation exposure. The technology that enables healthcare providers to make rapid and informed decisions also carries the inherent risk of triggering radiation-related health concerns, including cancer development [3,4].

In 1996, the International Commission on Radiological Protection (ICRP) introduced the term 'diagnostic reference level' (DRL). The reference level represents the level of risk above which it is generally judged to be inappropriate for planning to allow exposure to occur. The DRL has proven to be an effective tool for optimizing patient exposure during diagnostic and interventional procedures. The Commission considers that a dose rising to 100 mSv will almost always justify protective measures. However, Committee 3 advice does not specify quantities, numerical values, or details of implementation for DRLs. This is the task of regional, national, and local authorized bodies, each of which should meet the needs of their respective areas [5].

Several studies have highlighted the deterministic effects on thoracic organs [6], especially from low-dose radiation based on Japanese atomic bomb survivors who received a single, whole-body exposure to a range of doses of <5 Gy [7]. There are excess risks of heart disease for patients administered radiotherapy with estimated average heart doses of 1–2 Gy (single dose equivalent after correction for dose fractionation effects). Excess risk

of cardiovascular disease only becomes apparent 10–20 years after exposure to low doses. In a multivariable analysis, cardiac irradiation with individual total fractionated doses of 15 Gy significantly increased the hazard ratio of congestive heart failure (15–35 Gy, 2.2; >35 Gy, 4.5), myocardial infarction (15–35 Gy, 2.4; >35 Gy, 3.6), pericardial disease (15–35 Gy: 2.2; >35 Gy: 4.8), and valvular abnormalities (15–35 Gy: 3.3; >35 Gy: 5.5) compared with non-irradiated cancer survivors [8].

The median cumulative effective dose refers to the total amount of ionizing radiation received by a person over a period of time. Notably, repeated CT scans have subjected millions of patients to high cumulative radiation doses exceeding 100 mSv within 1 to 5 years, sparking discussions on the true risk levels associated with CT scans traditionally considered as low-dose. Data show that 0.6–4% of patients exceeded the 100 mSv threshold, underscoring the need for improved management and awareness of radiation exposure in medical settings [9].

This review delves deeply into the intricate relationship between radiological imaging and radiation exposure in ICUs. We examined the radiation exposure faced by COVID-19 patients in ICUs [10,11], the risks associated with repetitive imaging in ICU patients [12], and various degrees of radiation exposure in different patient populations [13].

The findings of this review emphasize the need for a meticulous approach to radiological practice, guided by the principles of justification, optimization, and strict adherence to dose limits, as advocated by the International Commission on Radiological Protection [14,15].

Striking the delicate equilibrium between the benefits of diagnostic imaging and the potential hazards of radiation exposure is a continuous challenge, particularly when caring for critically ill adults.

### 1.1. Life Span Study

The Life Span Study (LSS) of the Atomic Bomb Survivor Cohort, established in the 1950s, investigated the long-term health effects of atomic bomb radiation on approximately 120,000 survivors from Hiroshima and Nagasaki, including those exposed in utero and post-exposure-conceived children. This study monitors health outcomes, mortality, and cancer incidence. Findings indicate a heightened risk for leukemia and solid cancers, with leukemia rates increasing two years post-bombing and peaking after 6–8 years, while solid cancer risks grew by 40–50% per Gy among those exposed at 30 years old. Despite elevated risks in survivors, no increased risks were found in children born to exposed parents. This extensive research contributes to global radiation risk assessment, medical care, and policy formulation for radiation protection, supported by international collaboration and funding from Japanese and U.S. agencies, showcasing the collective effort to address the impact of radiation on human health [16].

### 1.2. Neuro-Intensive Care Unit

Neuro-Intensive Care Units (neuro-ICUs), established around 25 years ago, provide specialized care for neurological conditions like strokes, brain injuries, and CNS infections. Staffed with experts trained in advanced diagnostics and patient care, neuro-ICUs address the complex needs of patients with nervous system damage, aiming to improve their ability to move and communicate. Neuro-ICUs, however, face challenges with patients post neurosurgery, who may develop pulmonary complications, necessitating frequent radiological exams that contribute to significant cumulative radiation doses [17,18].

## 2. Methodology

We searched PubMed [19], Scopus [20], and Web of Science [21] for articles published between 1995 and 2023 about radiation exposure in ICUs.

The search parameters were "Cumulative Radiation Exposure", "X-ray", "Oncogenic", "Intensive Care Unit", "Critically ill patients", "Risk", and "Cancer", and the search was limited to articles in English. The final review excluded duplicate studies and articles that

did not specifically address radiation exposure in an ICU. We included evidence from case studies, editorials, observational studies, and randomized controlled trials (Table 1).

**Table 1.** Intensive care unit studies.

| | | |
|---|---|---|
| Radiation Exposure During the COVID-19 | Hadid-Beurrier et al. Radiat Res. 2022 [22] | • COVID-19 ICU patients received more radiological exams and had a higher CED of radiation than non-COVID-19 patients, indicating a need for radiation-reduction strategies. |
| The Risk of Repetitive Imaging in ICU Patients | Leppek et al. Radiologe 1998 [23] | • The risk from repeated chest X-rays in ICU patients with ARDS is minimal.<br>• Higher-than-recommended ICU radiation doses but a low overall estimated cancer risk due to this exposure. |
| | Slovis et al. Emergadiol 2016 [24] | • Slovis et al. reported higher-than-guideline ICU radiation doses with a low estimated cancer risk from CT scans, indicating a balance with patients' critical conditions. |
| | McEvoy et al. Crit. Care Resusc. 2019 [25] | • ICU patients' radiation exposure found most received low CEDs despite numerous procedures, with only a small percentage exceeding the high-risk threshold, indicating a generally low risk to patients. |
| | Rayo et al. J. Am. Coll. Radiol. 2014 [26] | • Rayo et al.'s study showed a 37% drop in abdominal CT scans and a 30–52% reduction in radiation exposure at a medical center, with a 63% decrease in projected radiation-induced cancers due to new protocols and technologies. |
| Radiation Safety Thresholds in ICU Patients | Krishnan et al. Chest 2018 [27] | • Krishnan et al.'s study found that 3% of MICU patients exceeded 50 mSv of radiation, and 1% surpassed 100 mSv, with CT and interventional procedures being the major contributors, highlighting the need for optimized radiological care. |
| | Rohner et al. Chest 2013 [28] | • Rohner et al. highlighted that ICU trauma patients are at risk of exceeding radiation limits, with 6.8% over 50 mSv, emphasizing the need for cautious use of diagnostic imaging. |
| Trends in Radiation Exposure in ICU Trauma Patients | Yee et al. Eur. J. Emerg. Med. 2012 [29] | • Yee et al. found increased radiation exposure in ICU trauma patients over time, with the median CED rising from 34.59 mSv in 2004 to 40.51 mSv in 2009, alongside more CT scans, highlighting a need for vigilant radiation monitoring. |
| | Moloney et al. World J. Radiol. 2016 [30] | • Moloney et al.'s study shows that CT scans, 16% of ICU imaging, contribute 97% to radiation doses, with higher doses linked to longer stays and trauma, indicating a need for CED reduction, particularly for younger patients. |
| Radiation Protocols in Neuro ICU Surveillance | Corcuera-Solano et al. AJNR Am. J. Neuroradiol. 2014 [31] | • Authors evaluated ultra-low-dose CT for NICU patient surveillance, finding that it significantly reduces radiation exposure while maintaining adequate image quality compared to standard protocols. |
| Radiation Exposure in Neuro ICU Patients | Chan et al. Neurohospitalist 2015 [32] | • Chan et al. investigated radiation exposure and tumor risks in ICU patients' heads and necks.<br>• Despite efforts to educate and inform, physicians' ordering habits persisted, indicating more action is needed to lower unnecessary radiation. |

**Table 1.** *Cont.*

| Measurement of Skin Dose from Neurological Imaging | Nawfel et al. AJNR Am. J. Neuroradiol. 2017 [33] | • Radiation from multiple CT/CTA scans in NICU patients may cause skin injury. <br> • A study with 52 patients indicates actual skin doses are lower than CTDI_vol estimates, yet repeated scans risk exceeding injury thresholds. |
|---|---|---|

## 3. Radiation Dose Assessment in ICU Patients Amidst COVID-19 and Beyond

### 3.1. Radiation Exposure during the COVID-19 Pandemic

Hadid-Beurrier et al. [22] conducted a study to assess radiation doses from medical imaging in COVID-19 ICU patients compared to non-COVID-19 critically ill patients. They performed a descriptive cohort study on 90 successive ICU patients with COVID-19 between March and May 2020, and 90 non-COVID-19 ICU patients from the same month in 2019. The CED was calculated from all radiological examinations. The authors found that COVID-19 patients underwent more radiological examinations, with a median of 12.0 mSv, compared to 4.0 mSv in non-COVID-19 patients. The CED over four months was significantly higher for COVID-19 patients at 4.2 mSv versus 1.2 mSv for non-COVID-19 patients. Among the survivors, a higher proportion of COVID-19 patients had a CED greater than 1 mSv. There was a significant correlation between the CED, the length of hospitalization, and the number of radiographic examinations. The authors found that critically ill COVID-19 patients underwent more medical imaging and thus had a higher CED than non-COVID-19 ICU patients, suggesting the need for strategies to reduce radiation exposure in the future [22].

### 3.2. The Risk of Repetitive Imaging in ICU Patients

Recent studies have examined radiation exposure risks in ICU patients. Leppek et al. [23] evaluated the morbidity risk associated with repeated bedside chest radiography in ICU patients, particularly those undergoing long-term ventilation for Adult Respiratory Distress Syndrome (ARDS) [13].

Body surface and gonadal doses were measured for each patient, revealing that the mean body surface dose per patient ranged from 0.31 mGy to 0.56 mGy. Gonadal exposure was less than 0.03 mGy per radiograph, with a mean effective dose per exposure of approximately 0.15 mSv.

The entrance surface dose (ESD) measures the radiation absorbed by the skin in mGy, typically using thermoluminescent dosimeters. It is a key benchmark in radiography for quality control and for setting diagnostic reference levels (DRLs) to ensure adherence to radiation protection principles [34].

The CED varied between 2.49 mSv and 14.09 mSv, corresponding to an estimated increase in individual cancer risk of between 0.01% and 0.07%. The study concluded that when considering the poor prognosis of critically ill, long-term ventilated patients, the additional risk of morbidity due to bedside chest radiographs is minimal and should be considered negligible.

Slovis et al. [24] focused on radiation exposure from CT scans in ICU patients and the associated lifetime attributable risk (LAR) of cancer. Conducted via an electronic chart review from January 2007 to December 2011, this study calculated the CED for each CT scan and predicted the LAR for each patient. The average radiation exposure was 22.2 mSv with a mean LAR of 0.1%, although the median was higher at 0.6%, ranging from less than 0.001% to 3.4%. The research found that radiation doses in the ICU were higher than those recommended by the guidelines, which could justify the critical condition of the patients. Overall, the estimated cancer risk due to radiation exposure in the ICU patient cohort was considered low [24].

McEvoy et al. [25] investigated CED exposure in ICU patients, considering the medical benefits against associated risks. Conducted as a retrospective audit in a South Australian tertiary care ICU, the study included 526 long-stay patients over a year from April 2015 to 2016. The audit revealed that these patients underwent 4331 procedures, resulting in a total of 5688.45 mSv of radiation. Despite constituting 82% of the procedures, chest X-rays contributed to only 1.2% of the CED. In contrast, although only 3.6% of the procedures were performed, abdominal and pelvic CT scans accounted for 68% of the CED. Over half of the patients received a CED of less than one mSv, 6% received >50 mSv, and 1.3% exceeded 100 mSv. Patients with trauma and longer ICU stays had higher CEDs. This conclusion indicates that most ICU patients receive a low CED, with the majority staying under 1 mSv, suggesting that the risk to patients is relatively low. These findings serve to inform clinicians about radiation exposure levels in ICU settings [25].

Rayo et al. [26] evaluated the impact of a reduced CT scan volume and dose reduction strategies on radiation exposure in patients at a Midwestern academic medical center from 2008 to 2012. The focus was on CT scans of the abdomen, head, sinus, and lumbar spine. Data collected for general medicine and ICU patients were used to assess the CT volume, rate, effective dose, radiation exposure, and estimated cancer risk annually. The results showed a significant 37% reduction in abdominal CT scan volume and a 30–52% decrease in radiation exposure due to dose-reduction strategies. No volume reduction was observed in the head or lumbar spine CT scans, and only a minimal decrease was observed in sinus scans. The combined strategies led to a 63% reduction in the estimated number of radiation-induced cancers. This conclusion emphasizes that the institution successfully reduced ionizing radiation exposure through fewer CT procedures and lower doses per procedure, primarily owing to the adoption of new protocols and technologies. These changes appear to have had the most substantial impact on reducing the future cancer risk associated with CT radiation [26].

These studies collectively underline the importance of balancing medical benefits against the potential risks of radiation in critical care settings.

## 4. Long-Term Studies on Radiation Exposure in ICU Settings
### 4.1. Radiation Safety Thresholds in ICU Patients

The implications of radiation exposure in critical care settings have been a growing concern, particularly as doses may approach or exceed established safety limits.

Krishnan et al. [27] evaluated patients' radiation exposure in the medical ICU (MICU), hypothesizing that some may exceed US federal occupational health standards. This retrospective observational study at an academic medical center analyzed all adult admissions to the MICU in 2013, totaling 4155 patients, to calculate their CED from radiological studies. The results showed that 3% of admissions accrued a CED of ≥50 mSv, and 1% exceeded 100 mSv, with a median CED of 0.72 mSv. Higher APACHE III scores, longer MICU stays, sepsis, and gastrointestinal issues were associated with higher CEDs. CT and interventional radiology procedures were the most significant contributors to the CED. The study concluded that a notable proportion of MICU patients receive radiation doses exceeding 50 mSv, with some surpassing 100 mSv. There is a clear need to justify and optimize radiological studies to minimize exposure while delivering essential medical care.

Rohner et al. [28] discussed the concern regarding the safety limits of radiation exposure to minimize the risk of radiation-induced cancer. Occupational exposure limits were set at 20 mSv/year over five years, with a cap of 50 mSv per year. However, the average radiation dose in the US has increased over the past 30 years, mainly because of medical imaging. The study hypothesized that patients in a surgical ICU, particularly trauma patients, might approach or exceed these exposure limits owing to frequent diagnostic imaging. This study involved prospective observations of patients in a level I trauma center SICU over 30 days. Radiation doses were calculated using Huda's method for all imaging procedures. The study found that 6.8% of patients exceeded the 50 mSv mark. Higher radiation doses were associated with trauma, extended hospital stays, and more frequent

use of CT and fluoroscopy. Multivariable analysis indicated that the number of CT scans and fluoroscopy duration significantly influenced an increased radiation exposure. This finding emphasizes the need for careful radiological imaging to avoid excessive radiation exposure. Healthcare providers must balance the diagnostic benefits with the risks of the CED in critically ill and injured patients [28].

### 4.2. Trends in Radiation Exposure in ICU Trauma Patients

Yee et al. [29] assessed the CED of radiation received by mechanically ventilated trauma patients in the emergency department and ICU during two periods. A retrospective analysis was conducted on two cohorts of 45 adult patients each, starting from 1 January 2004 and 1 January 2009 in a regional non-urban ICU. Data on radiological examinations, demographics, and clinical information were collected from various databases. The findings suggested an increase in the median CED per patient from 34.59 mSv (IQR 9.08–43.91) in 2004 to 40.51 mSv (IQR 22.01–48.87) in 2009, with a significant p-value of 0.045. There was also an increase in CT examinations per patient over time, from an average of 2.11 in 2004 to 2.62 in 2009. The conclusion drawn was that radiation exposure in ICU trauma patients requiring mechanical ventilation has increased, emphasizing the need for prospective monitoring and awareness among staff of the heightened risk due to this shift in clinical practice [29].

Moloney et al. [30] quantified the CED of radiation from diagnostic imaging performed in ICU patients. Conducted prospectively in the ICU of a tertiary referral and level 1 trauma center, this study gathered demographic and clinical data from all patients admitted for over one year. The CED was calculated based on the UK National Radiation Protection Board's reference effective dose. Of the 421 patients, 2737 studies were conducted, resulting in a total CED of 1704 mSv. The median CED was 1.5 mSv. In the pediatric subgroup, the total CED was 74.6 mSv, with a median of 0.07 mSv. Chest radiography, although the most common, accounted for only 2.7% of the total CED, while CT scans, only 16% of the studies, contributed to 97% of the CED. Patients with trauma had a significantly higher CED than those with medical or surgical trauma. This study found that the length of ICU stay was an independent predictor of receiving a CED > 15 mSv. This conclusion highlights that trauma patients and those with extended ICU stays are at higher risk of elevated CEDs. This study advocates minimization of the CED, particularly in younger patients [30]. These studies underscore the urgent need to balance the necessity of diagnostic imaging in critically ill patients with the potential long-term risks of radiation exposure.

## 5. Optimizing Radiological Practices in Neuro ICU Units
### 5.1. Evaluating Radiation Protocols in Neuro ICU Surveillance

Corcuera-Solano et al. [31] focused on the radiation exposure of patients in the neurosurgical ICU (NICU) who underwent multiple head CT scans. The objective of this study was to evaluate the effectiveness of an ultra-low-dose CT protocol for NICU surveillance, comparing it to standard low-dose CT and traditional standard-dose CT protocols. A retrospective analysis of 54 head CT examinations of 22 NICU patients was conducted. The examinations were categorized into ultra-low dose (22), low dose (12), and standard dose (20). The ultra-low-dose and low-dose CTs used a sinogram-affirmed iterative reconstruction technique on a Siemens AS + 128 scanner. In contrast, standard-dose CTs used filtered back-projection on a Somatom Sensation 64 scanner. Image quality and radiation dose were assessed both qualitatively and quantitatively. The results showed that ultra-low-dose CT had a 68% lower dose index volume than standard-dose CT but maintained similar image quality and signal-to-noise ratio (SNR). Low-dose CT had a better image quality than standard-dose CT, with a 24% lower dose index volume. Although ultra-low-dose CT had a lower SNR than low-dose CT, it is still clinically acceptable. The study concluded that the ultra-low-dose CT protocol significantly reduced radiation exposure while preserving an adequate image quality for surveillance in NICU patients [31].

*5.2. Radiation Exposure in Neuro ICU Patients*

Chan et al. [32] examined the CED from diagnostic studies in patients with primary neurological disorders in an ICU. This study aimed to quantify the radiation doses and assess the risk of radiation-induced tumors in the head and neck regions. This retrospective cohort study was conducted in a single institution's neuroscience ICU (NICU), with radiation doses converted to estimated effective doses in mSv using published formulas. An educational initiative was implemented to inform physicians about patient radiation exposure, with a focus on treating acute subarachnoid hemorrhages. Data on radiation exposure were posted at patients' bedsides to determine whether it would influence physician ordering practices. From July 2010 to March 2011, 641 patients who underwent head CT scans were identified, with an average exposure of 18.4 mSv. Patients with subarachnoid hemorrhages had the highest average exposure at 37.1 mSv, although the risk of carcinogenesis was deemed low. The educational initiative did not result in a reduction in the effective dose per patient. The study concluded that while it is possible to accurately report estimated effective doses to physicians, more than an educational initiative alone was needed to alter ordering behaviors. These findings suggest that additional strategies are necessary to mitigate unnecessary radiation exposure in the NICU [32].

*5.3. Direct Measurement of Skin Dose from Neurological Imaging*

Nawfel's research [33] addresses the concerns that radiation exposure from multiple CT and CTA scans in neurological ICU (NICU) patients can reach levels high enough to cause deterministic skin injuries. This study aimed to measure head CT and CTA peak skin doses directly, evaluate their correlation with the volumetric CT dose index (CTDI_vol), and assess whether the CED from multiple scans could exceed the threshold for skin injury. From 2011 to 2013, a prospective study involving 52 patients measured peak skin doses using nanoDot optically stimulated luminescence dosimeters across two CT scanners. Patient and phantom data were collected to ensure accuracy. The study found that CTDI_vol often overestimated the peak skin dose by 1.4- or 1.9-fold. The CED for patients who underwent multiple scans ranged from 1.9 to 4.5 Gy. In conclusion, the directly measured skin doses from head CT and CTA scans were lower than those estimated by CTDI_vol. However, the CED from multiple examinations may surpass the deterministic threshold for skin damage in NICU patients, indicating the potential risk of injury from repeated imaging [33].

## 6. Discussion

This review highlights concerns regarding the risks of radiation exposure from frequent CT scans in oncological patients and underscores the essential nature of these examinations. Lencioni et al. investigated the appropriateness of follow-up CTs against AIOM guidelines in a large oncology patient sample and revealed that most scans adhere to these guidelines. This adherence suggests that for many patients, the benefits of timely detection and monitoring of cancer recurrence, metastases, or new tumors significantly outweigh the potential late side effects of radiation. Therefore, it is crucial to balance the need for vigilant follow-up with the risk of unnecessary exposure [35]. It is also essential to provide dermatologic and echocardiographic follow-up in patients who have had a long ICU stay burdened with numerous radiologic investigations (particularly chest radiographs and CT scans) because of the risk of deterministic effects, even with a long latency.

Acknowledging the current strategies employed to mitigate exposure to ionizing radiation remains the cornerstone of patient safety in ICU settings. New dose-reduction strategies are mainly employed during procedures such as percutaneous coronary interventions. One noteworthy approach is the use of "protective drapes" during these procedures. These drapes are placed over patients to shield them from scattered radiation, effectively reducing the dose received by patients and staff [36].

In addition, novel imaging technology photon-counting CT (PCCT) scanners can reduce radiation doses and improve the image quality. PCCT allows for better signal-

to-noise ratios (SNRs) for the same dose, especially in overweight patients, resulting in images of comparable quality to traditional CT [37]. The implementation of dose-tracking software and real-time dose monitoring provides immediate feedback, facilitating dose optimization. Moreover, educational programs aimed at raising awareness among healthcare providers about the principles of radiation safety—justification, optimization, and limitation—are instrumental in cultivating a mindful imaging culture [38]. These current methodologies demonstrate progress in radiological practices and highlight the potential for future innovations to reduce further risks associated with ionizing radiation.

This study had several limitations. It primarily relies on existing research that may be geographically limited, raising concerns about potential biases and the applicability of findings across diverse healthcare systems. Including studies with varied designs, such as case studies, observational studies, and trials, leads to noncomparability. The rapidly advancing field of medical imaging might need to be fully represented, particularly the latest advancements in radiation reduction, which are crucial in contemporary practice. The absence of long-term follow-up data limits insights into the prolonged effects of radiation exposure, which is particularly relevant for patients recovering from critical conditions such as COVID-19.

## 7. Conclusions

This review critically examines the challenges of managing radiation exposure in ICUs, particularly during the COVID-19 pandemic. It emphasizes the increased reliance on imaging for critical care diagnostics and the consequent rise in radiation doses for ICU patients. This trend raises significant concerns about the long-term risk of cancer, especially in vulnerable groups such as trauma patients. The review highlights the importance of meticulous radiation management, focusing on dose minimization and strict adherence to safety thresholds.

Key strategies for optimizing radiological practices in ICUs include the implementation of ultra-low-dose CT protocols, especially in neuro-ICUs, and the direct measurement of skin doses. Educational initiatives to increase awareness about radiation risks among healthcare providers are also essential. These measures, along with protective drapes during procedures and real-time dose monitoring systems, represent a shift towards reducing radiation exposure without compromising diagnostic quality.

It is also important to provide dermatologic and echocardiographic follow-up in patients who have had a long ICU stay because of the risk of deterministic effects, even with a long latency.

In conclusion, we emphasize the importance of a balanced approach to radiological imaging in ICUs, stressing the need to minimize radiation exposure, while ensuring the continuity of high-quality medical care. This approach involves a combination of innovative dose-reduction strategies, optimized imaging protocols, and educational efforts to manage radiation risks effectively.

**Author Contributions:** Conceptualization, C.Z. and E.Q.; methodology, C.Z.; software, A.T.; validation, C.Z., A.T., C.B. and T.B.; formal analysis, C.Z.; investigation, A.T.; resources, T.B.; data curation, C.B.; writing—original draft preparation, C.Z.; writing—review and editing, E.Q.; visualization, C.Z.; supervision, E.Q.; project administration, E.Q. All authors have read and agreed to the published version of the manuscript.

**Funding:** This research received no external funding.

**Institutional Review Board Statement:** Not applicable.

**Informed Consent Statement:** Not applicable.

**Data Availability Statement:** Data sharing is not applicable to this article.

**Conflicts of Interest:** The authors declare no conflicts of interest.

## Abbreviations

| | |
|---|---|
| ABCC | Atomic Bomb Casualty Commission |
| APACHE III | Acute Physiology and Chronic Health Evaluation III |
| ARDS | Adult Respiratory Distress Syndrome |
| CED | Cumulative Effective Dose |
| CT | Computed Tomography |
| CTDI_vol | Computed Tomography Dose Index Volume |
| DRL | Diagnostic Reference Level |
| IAEA | International Atomic Energy Agency |
| ICRP | International Commission on Radiological Protection |
| ICU | Intensive Care Unit |
| LAR | Lifetime Attributable Risk |
| LSS | Life Span Study |
| MICU | Medical Intensive Care Unit |
| NICU | Neurosurgical or Neuroscience Intensive Care Unit |

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
