# Peer review of "Radiation Overuse in Intensive Care Units"

_tomography, doi:10.3390/tomography10020015_

Round 1

Reviewer 1 Report

Comments and Suggestions for Authors

1.      The technology that enables healthcare 49 providers to make rapid and informed decisions also carries the inherent risk of triggering 50 radiation-related health concerns, including cancer development.

Please add that there are data that show what doses of x-rays are critical for particular organs, for example, in the chest. We have data based on oncology patients e.g. with breast cancer who experience radiation therapy. They are coronary artery lesions, pericarditis or degeneration of heart valves.

2.      The data show an important but little-noticed problem. Please point out in conclusion that if tests like follow-up CTs are necessary to be performed, this outweighs the possible late side effects. But it should be pointed out by physicians discharging such a patient home that follow-up in a dermatology clinic or control echo studies are necessary in the case of a chest CT.

Reviewer 2 Report

Comments and Suggestions for Authors

Chiara Zanonet al. present a review article to highlight the essential but risky role of radiological imaging in critical care. The methodology is correct and consistent conclusions are related to the evidence and arguments presented. The table is well presented.
Although the topic is interesting and the manuscript well-written, some considerations need to be clarified.

1. Introduction: Add additional details on the field of discussion that will be highlighted in the manuscript (e.g. Neuro ICU Units).

2. Introduction: Add more details about the concept of “median Cumulative Effective Dose”.

3. Please, add a paragraph before the conclusion on “Gaps in evidence and future perspectives”. This paragraph highlights the current way to reduce the exposition to ionizing radiation. Discuss the interesting modality used during percutaneous coronary procedures with the role of adjunctive protective drapes.

Reviewer 3 Report

Comments and Suggestions for Authors

This review manuscript includes useful information on the radiation exposures in ICU. However, this is not sufficient as a review and revisions and additional descriptions would be required.

Overall

This manuscript consists of Introduction, Methodology, each paper contents and Conclusion. There is no Discussion nor any analysis based on the contents of papers. Though this is not a paper with original data but a review of the previous papers, some discussions and analyses of the data, and the limitation of this report etc. would be necessary.

Key points:

L28

The meaning of “recommended radiation safety thresholds” is unclear. For medical exposures of the patients, justification and optimization are considered, but there is no dose limit. If the author would like to mention this issue, they should be clearly and correctly described.

1.Introduction

L57

According to the ICRP 103, no dose limit is applied for the patients as commented above. They should also be mentioned.

3. Radiation Dose Assessment in ICU Patients Amidst COVID-19 and Beyond

L79

There is no unit for 12.0 and 4.0. They should be written with the unit mSv.

L117

It is not appropriate to compare radiation exposures of the patients and the public exposure limit of 1mSv regarding radiation protection system by the ICRP though this is referred from a paper by McEvoy et al.

4.Long-Term Studies on Radiation Exposure in ICU Settings

L 148

As well as L117, the comparison with radiation doses of the patients and the annual occupational limits is not appropriate.

5.3 Direct Measurement of Skin Dose from Neurological Imaging

L240

1.9 4.5 -> 1.9-4.5

Title and Coclusion

The meaning of “a Cause for Concern” is not clear in this manuscript. The word “cause” should be excluded in the title, otherwise the descriptions based on the references should be included in the text.

Reviewer 4 Report

Comments and Suggestions for Authors

DEAR AUTHORS,

The Manuscript is very important and interesting for Tomography journal readers.  

Specific comments:

Abstract:

Explain the abbreviation ICU only when it first appears in the text.

intensive care unit or Intensive Care Unit?

 Introduction

Line 48: change

 monitoring. [1]–[3]

to

Monitoring  [1]–[3].

Line 51: change

development. [2], [4], [5] to

development  [2], [4], [5].

Apply throughout the paper.

For dose limits please cite  the appropriate International Commission on Radiological Protection (ICRP). Change ref 12.

Methodology

Pp4, line 91: explain “surface dose”

Pp4, line 140:

add the reference number

Krishnan et al. [22]

pp. 6 line 231: add the reference number after “ Nawfel's research”

Apply throughout the paper.

Pp. 6, line 240: correct

ranged from 1.9 4.5 Gy

Conclusion

Please improve the conclusion section.

Round 2

Reviewer 3 Report

Comments and Suggestions for Authors

The author revised the manuscript as the reviewer previously commented. It is judged that this manuscript is acceptable.